

# Virtual fieldtrip to the Esla Nappe (Cantabrian Zone, NW Spain): delivering traditional geological mapping skills remotely using real data

Manuel I. de Paz-Álvarez[1], Thomas G. Blenkinsop[2], David M. Buchs[2], George E. Gibbons[2], Lesley Cherns[2]

[1] Departamento de Geología, Universidad de Oviedo, C/ Jesús Arias de Velasco s/n, 33005 Oviedo, Spain
[2] School of Earth and Environmental Sciences, Cardiff University, Main Building – Park Place, CF10 3AT, Cardiff, UK

*Correspondence to*: Manuel I. de Paz-Álvarez (pazmanuel@uniovi.es)

**Abstract.** The restrictions implemented to contain the spread of the COVID-19 pandemic during 2020 and 2021 have forced
university-level educators from around the world to seek alternatives to residential physical field trips which constitute a fundamental pillar of geoscience programmes. The field-mapping course for 2nd year Geology BSc students from the Cardiff University was replaced with a virtual mapping course set in the same area as previous years, the Esla Nappe (Cantabrian Zone, NW Spain). The course was designed with the aim of providing the students with the same methodology employed in physical mapping, gathering discrete data in stops located along five daily itineraries. Data included bedding attitude, outcrop
descriptions with a certain degree of ambiguity, photographs and/or sketches, panoramic photos and fossil images. Data was provided to the students through georeferenced KMZ files in Google Earth. Students were asked to keep a field notebook, define lithological units of mappable scale, identify large structures such as thrust faults and folds with the aid of age estimations from fossils, construct a geological map on a hard-copy topographic map, draw a stratigraphic column and cross sections, and plot the data in a stereonet to perform structural analysis. The exercise allowed a successful training of diverse
geological field skills.  In the light of the assessment of reports and student surveys, a series of improvements for the future is considered. Though incapable of replacing a physical field course, the virtual exercise could be used in preparation for the residential fieldtrip.

## 1 Introduction

Fieldwork has traditionally been an integral part of Geoscience teaching programs, and of Geology in particular (Compton,
1958; Boyle et al., 2007; Butler, 2008; Mogk and Goodwin, 2012). The benefits of fieldwork teaching are varied, and include an appraisal of theoretical concepts through practice and sensorial experiences (Gibson, 2007), the learning of practical geological skills needed by researchers and practitioners (Butler, 2008; Whitmeyer et al., 2009), an increase in student engagement and enjoyment (Boyle et al. 2007; McConnel and van der Hoeven Kraft, 2011), as well as the development of transversal competencies such as teamwork, communication, critical thinking and autodidacticism (Arrowsmith et al., 2011).



Fieldwork relevance is recognised by bodies that regulate geology degree programmes, which either establish a minimum
      curricular content (e.g. 60 days for 3-year degrees in the UK; Geological Society of London, 2013), or tacitly consider it an
      essential curricular content (e.g. 40 – 80 days for 4-year degrees in different Spanish universities).

      Fieldwork has traditionally involved different pedagogical methods. Among them, one of the most traditional consists of
      delivering a lecture in a specific location in the field, while students take notes, observe the explained phenomenon, and attempt

to memorise it. This generates an enhanced passive learning environment where the students may gain knowledge thanks
      solely to the lecturer's expertise. However, it is well known among the cognitive science community that the knowledge gained
      through passive learning is more superficial than that gained through active learning, where students are involved in the
      learning process through practise and the undertaking of lecturer-designed activities (e.g. Higgs and McCarthy, 2005; Moran,
      2018). While the benefits of fieldwork have long been assumed by educators, it has been difficult to demonstrate them clearly

(e.g. Elkins and Elkins, 2007; Stokes and Boyle, 2009), a proviso that applies even more to virtual fieldwork.

      Circumstances can limit the field access in the learning environment. Although exceptional, these situations have proved to be
      very trying to both staff and students, as substitute activities need to be implemented, usually in a limited time interval. This
      was the case, for example, during the outbreak of foot and mouth disease in 2001, when travel restrictions were imposed to
      travelling outside the UK, thus limiting curricular fieldwork activities (Placing and Fernandez, 2002), or during the 2010-2011

Canterbury earthquake swarm in New Zealand, which prompted a rapid substitution of college lectures with virtual lessons
      (Mackey et al., 2012). The ongoing coronavirus pandemic (COVID-19) has caused a major disturbance in education worldwide
      (Sahu, 2020), especially in highly practical disciplines where fieldwork and face-to-face practicals are essential curricular
      activities (e.g. Ferrel and Ryan, 2020; Day et al., 2021; Rotzien et al., 2021). In these circumstances, many lectures have been
      moved to the virtual environment, where social interactions are reduced and the spread of the disease is minimised (e.g. Sahu,

2020). While this transition may be challenging for theory lessons, it is especially arduous for practicals, and in particular for
      fieldwork (Bryson and Andres, 2020; Day et al., 2021; Rotzien et al., 2021).

      Virtual fieldwork has then come to the rescue of these particular curricular formative activities, although it has been in use
      long before the COVID-19 crisis. Virtual fieldwork refers to the recreation of field-based activities in a classroom environment,
      most commonly aided by technology (Maskall and Stokes, 2008). The advantages of virtual fieldwork include, among others:

(i) an absence of field-related logistical issues, (ii) its accessibility to differently abled students and to those unable to attend a
      physical trip for other reasons, (iii) flexibility of teaching resources, (iv) ease of repetition and adaption of the exercises as
      many times as necessary, (iv) ability to incorporate and interact with non-outcrop features such as geochemical data or seismic
      lines, (v) ability to use a variety of observation scales, (vi) training of field skills prior to physical fieldwork, and (vii)
      opportunity to visit, even if virtually, locations that otherwise would be difficult, and in some cases impossible, to access

(Hurst, 1998; Stokes et al., 2012; Cliffe, 2017).

      Just like real fieldwork, its virtual counterpart has many different possible delivery modes, all of them with advantages and
      drawbacks. Among them, there are (i) virtual video fieldtrips, recorded lectures based on field information and photographs;
      (ii) text-book supplements with field case studies with data and related exercises (Ford and Hipple, 1999); (iii) virtual lab



experiences where a lab environment is simulated in order to learn basics of a certain subject, such as fossil identification

(Placing and Fernandez, 2002); (iv) adaptive e-learning exercises where feedback is provided to students based on their responses and decisions (e.g. Mead et al., 2019); (v) virtual educational games where students use an avatar to interact with different information and scenarios where a certain learning task needs to be performed (e.g. Pringle, 2013, Virtual Landscapes, 2020; Needle et al., 2021); (vi) virtual reality tours which provide the students with an immersive 3D experience using a head mounted display, through which they may interact with a virtual environment and utilise different data; (vii) virtual globe

experiences, where a 3D visualisation of rocks, structures, and topography is possible, which may be supplemented with photographs, sketches and outcrop descriptions (Blenkinsop, 2012; Whitmeyer, 2012; Whitmeyer and Dordevic, 2020; Rotzien et al., 2021; Bond and Cawood, 2021; Toy et al., 2021; Mahan et al., 2021; Marshall and Higley 2021). The effectiveness of these trainings depends largely on the degree of student engagement in the learning process itself, the type of information available to the students, the way in which the working data is provided, and the nature of the final outcome of the activity. In

many cases, the virtual fieldwork is little more than a 'show and tell' activity in which students are presented with images and data with which they do not work with, thereby loosing much of the pedagogical potential that virtual fieldwork can provide. In this contribution, we present a virtual fieldwork course, the Esla Mapping Project (EMP), designed during the spring of 2020 for second year Geology BSc students of the School of Earth and Environmental Sciences of Cardiff University. The virtual fieldwork was produced as a substitute mapping exercise for students in the academic years 2019/2020 and 2020/2021

as a consequence of the COVID-19-related restrictions in place, that resulted in the closure of the campus and the generalised lockdown in the UK and elsewhere. The EMP was designed to achieve as many of the aims as possible of the original field-based mapping, including a strong emphasis on development and testing of hypotheses, development of 3 and 4D thinking, and understanding of how structural and stratigraphic knowledge iteratively converge to a solution. The results reported in this contribution are largely referred to the 2019/2020 version of the EMP, which has been slightly improved during the second

2020/2021 run.



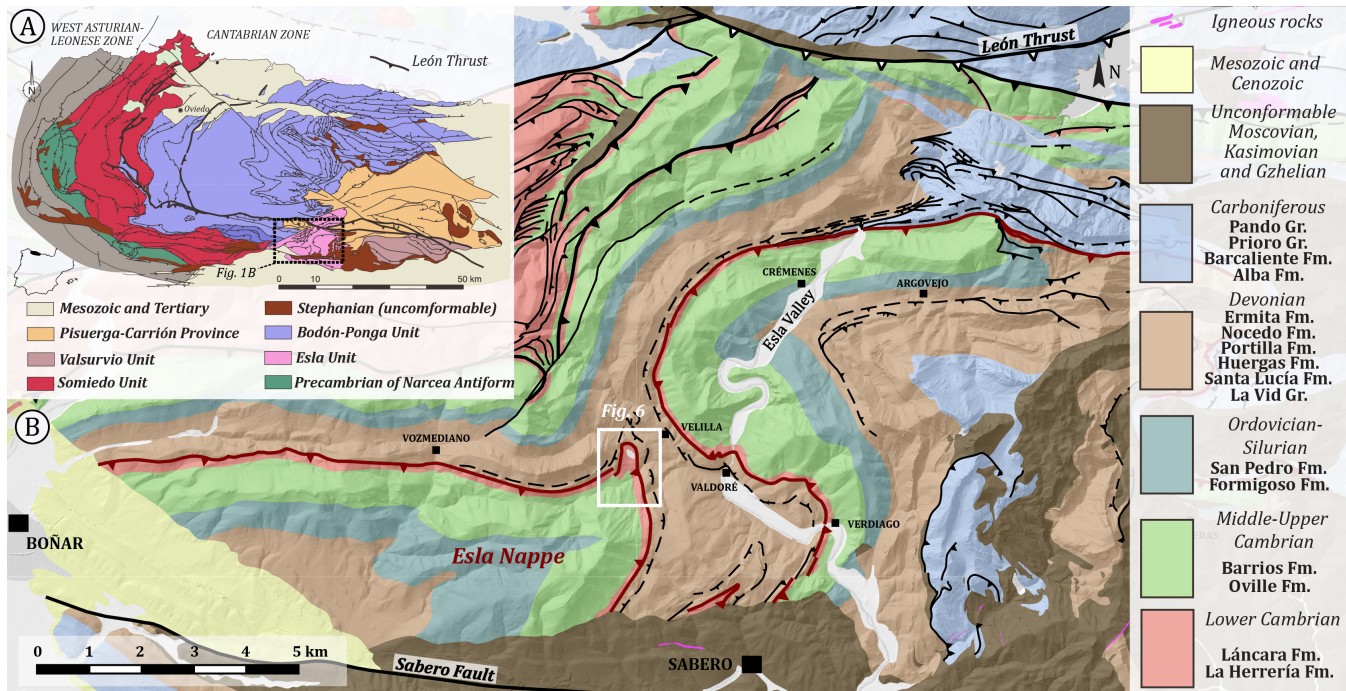

Figure 1. (a) Geological map of the Cantabrian Zone modified from Alonso *et al.* (2009). (b) Simplified geological map of the Esla Nappe modified from Merino-Tomé *et al.* (2014).

## 2 Design of the fieldtrip

### 2.1 Working area

In previous years, a five-day field mapping course was run in the Cantabrian Mountains of NW Spain for $2^{nd}$ year students of Cardiff University. The trip was centred in the Esla Nappe, one of the main thrust units that comprise the Cantabrian Zone, the foreland-fold-and-thrust belt of the Variscan Orogen in the NW Iberian Massif (Fig. 1; de Sitter, 1959; Rupke, 1965; Arboleya, 1978; Alonso, 1987; Alonso et al., 2009; and references therein). The major geological features of the mapping area are a sequence of Cambrian to Devonian carbonates and siliciclastic rocks deformed at sub greenschist facies in the Variscan orogeny. The major structure consists of a nappe (the Esla nappe) that carries Cambrian and Ordovician formations over Silurian-Devonian rocks along a bedding-parallel thrust in the mapping area. The nappe has been folded in two major different orientations, giving rise to a relatively complex outcrop pattern. The recognition of the thrust depends on the correct interpretation of the age sequence via fossil evidence, and is locally supported by breccia outcrops on or near the thrust plane. The area is exceptional for geological mapping training owing to the outstanding exposure, the variety of sedimentary rocks, the good preservation of thrust sheet geometry and structures, and the subsequent deformation of the nappe through a set of folds that result in a variety of cartographic patterns within a relatively small area.



The virtual Esla Mapping Project (EMP) was designed to cover the same area used in previous years, around the locality of La Velilla de Valdoré. The location selection was made based on three key motives: (i) using the same location would allow

for continuity in the learning of students across different years, (ii) the virtual exercise could be used in later years as an introduction to the real fieldtrip, and most importantly, (iii) the authors' detailed knowledge of the Esla Nappe from both a research and educational point of view.

## 2.2 Student background

The EMP was planned for 2nd year students of the 3–year Geology and Exploration Geology BSc degrees at Cardiff University,

a group of 60 students. At this stage the students have had limited field experience, including only basic mapping training. Students had, however, previously received training in the description and interpretation of structural, sedimentary, and palaeontological features adapted to the geological complexity of the mapping area. The residential fieldtrip to the Esla Nappe was intended to provide them with the necessary geological and organisational skills to undertake a five-week independent mapping project during the summer between year two and three, which eventually results in their Geology Dissertation,

presented in their 3rd year as a key part of their BSc degree.

**Figure 2. Outline of different elements of the Esla Mapping Project. (a) Daily itineraries of realistic walking distance with a number of stops along the way. X.Y notation refers to the day (X) and the stop (Y). Map derived from digital elevation model provided by IGN (National Geographic Institute of Spain). (b) Tilted view of the northern part of the mapping area, looking north, with the stops in white symbols (©Google Earth 2021). Based on the information contained on each of the stops, students coloured the pins accordingly. (c) Example of data provided to the students in one of the stops. The photo of the trilobite is accessed by a link to an image stored in the GitHub repository, opened within Google Earth (©Google Earth 2021). (d) Panoramic view provided in stop 3.13 looking west: students were asked to inspect it and draw an annotated field sketch. (e) Outcrop photograph provided in stop 4.3 to familiarise the students with the provided rock description. (f) Outcrop sketch provided as data to the students in stop 5.1.**

## 2.3 Fieldtrip organisation

The virtual fieldtrip was designed to mimic physical mapping methodology and experience as closely as possible. The ultimate goal of the exercise was for the students to be able to generate a geological map of a portion of the Esla Nappe remotely while at the same time getting a feel for some of the field-based procedures. The exercise involves an adapted conventional field methodology where the students keep a record of field observations, annotations and bedding attitude in a field notebook, taken in certain localities recorded on a topographic map. For this purpose, five realistic daily itineraries were designed with similar lengths to those covered during real fieldwork days, with between 10 and 17 localities per day (Fig. 2). A variety of





information is provided to the students in the form of field annotations for each of the stops, and included (i) field outcrop descriptions of the rock types, sedimentary structures and structural features, (ii) outcrop photographs, (iii) annotated outcrop field sketches, (iv) outcrop-scale stratigraphic columns, (v) fossil photographs and/or fossil species found in the locality, (vi)

panoramic photographs taken from the location, (vii) bedding attitude and, in some cases, way-up criteria (Fig. 2). The data is distributed among the different stops, so that not all of them contain the same type of information. In addition, all locations include their geographic coordinates in the WGS 84 coordinate system. All the provided data is a truthful representation of real locations in the Esla Nappe, except in the case of some fossils, which have been found in other areas of the Cantabrian Zone. With the exception of the fossil images and species, which were obtained from the literature (Supplementary material),

all the bedding measurements, field descriptions, photographs and sketches are original data collected by the authors throughout the years. The wealth of these data has been key to the successful implementation of the EMP.

**2.4 Materials used**

All the information was made available to the students in the form of a KMZ file to be opened in Google Earth (GE). The selection of GE as the platform was based on its ease of installation, its performance on average personal computers, and the

students' previous learning experience with the software, which has included a basic mapping exercise. Other geographic information systems were considered, such as QGis®, but were rejected due to their steeper learning curve for the students. Nonetheless, some of the material provided to the students was produced with the aid of QGis® software, and then exported as KMZ files to be used in GE. Each of the stops in the KMZ file contained a text description with the provided data, as well as a link to web-hosted material such as photographs or sketches. In order to minimise computing resources on the students'

varied range of computers, the decision was made to host the graphic material such as sketches and field photographs in a web application, instead of inserting them within the KMZ file, which could have resulted in an unsatisfactory GE performance in some cases. The chosen web application was GitHub repository, due to (i) its widespread use among the coding community, which ensures a long-term storage of the files; (ii) the fact that it is free for open-source projects; (iii) its good performance for repositories with less than 1 Gb; (iv) and an ample file size limit of 100 Mb, more than enough for photographs and sketches.

The repository generates readily accessible file links that can be inserted in the text in the KMZ files. The links redirect to the material from within GE, without the need to open third-party web-browsers.

In addition to this KMZ file with the outcrop locations, data and observations, students were also provided with topographic contours in a KMZ file with 5 m intervals generated in QGis® from a digital elevation model obtained from the IGN (National Geographic Institute of Spain), a SK2 spreadsheet to generate 3D structural symbols showing the orientation of planar and

linear features in GE (Blenkinsop, 2012), a fossil guide with identification keys and related ages based on specimens sampled in the area over the years, and a mapping guide. The mapping guide informed the students of the materials available to them for the completion of the EMP, the required software operation, the learning outcomes of the exercise, and instructions for the different tasks to be performed. In addition to the digital dataset, the students were provided with the physical materials necessary to complete the exercise. This material included (i) a hard copy of a topographic map with basic topographic





contours, elevation points, coordinates and northing and easting grid (Supplementary material); (ii) a hard copy of a
      stereographic projection net, tracing paper and pin, (iii) graph paper for the cross sections and iv) a field notebook. Owing to
      the lockdown in place, this material was delivered by post to all the students.

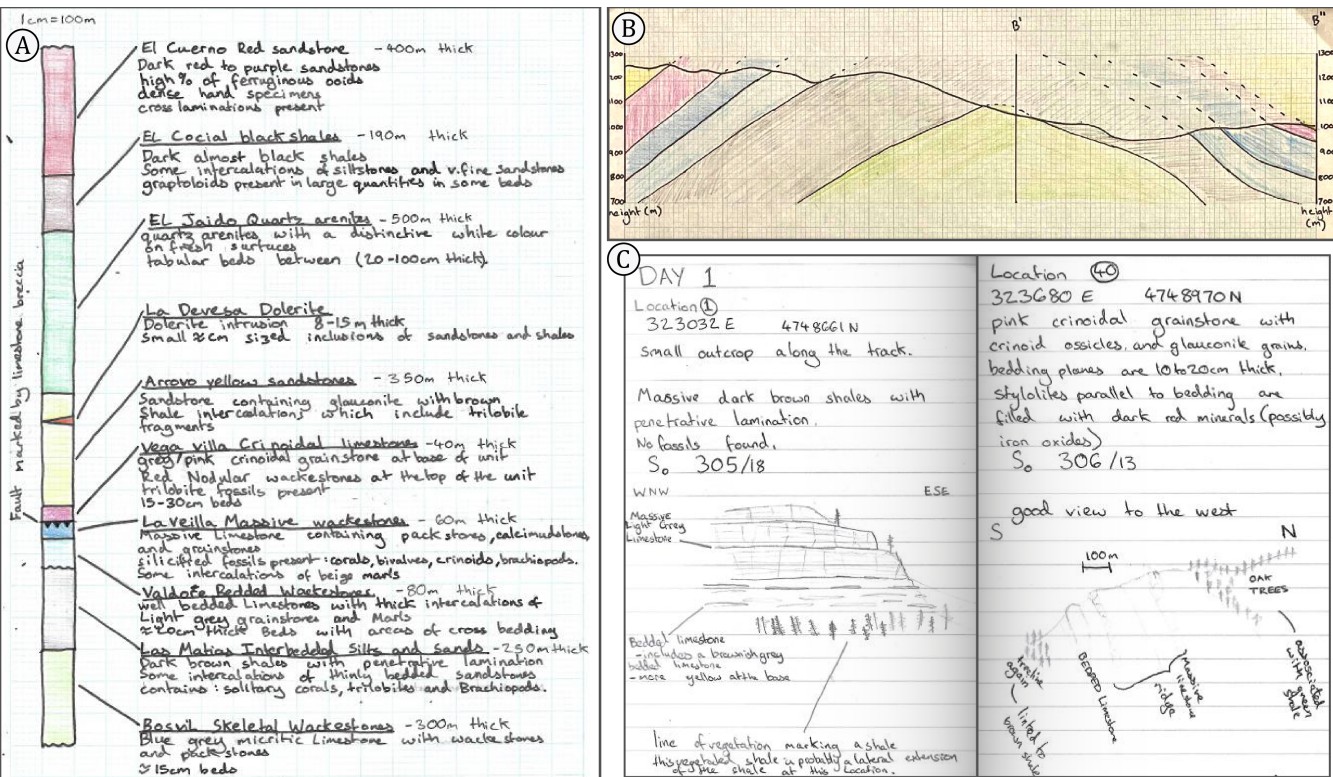

**Figure 3. Tasks performed by students during of the Esla Mapping Project. (a) Stratigraphic log of the lithological units identified**
**in the area, displayed in a tectono-stratigraphic order. (b) Cross-section through the central part of the mapping area. (c) Example**
      **of field notebook pages where a student has transcribed the information provided and drawn field sketches from panoramic**
      **photographs (see Fig. 2D for comparison with the lower-right sketch).**

**2.5 Learning outcomes**

      The field trip was designed to achieve the learning outcomes that are specified in Table 1. The table also shows the learning
outcomes created for virtual field experiences by the National Association of Geoscience Teachers (NAGT, 2020). These have
      similarities with some of the more specific learning outcomes of the EMP. The learning outcomes of the EMP are the same as
      those for the physical trip that was run in previous years: the obvious difference between the trips is that the list of skills
      expected from the physical trip includes practical aspects such as the use of a compass-clinometer, and locating oneself in the
      field on a topographic map.




| NAGT | Esla Mapping Project | Task involved |
|---|---|---|
| Collect accurate and sufficient data on field relationships and record these using disciplinary conventions (field notes, map symbols, etc.) | Recording the rock record in a field notebook | Daily summaries |
| Interpret earth systems and past/current/future processes using multiple lines of spatially distributed evidence | Using observations from sedimentary rocks to interpret sedimentary processes and depositional environments | Daily summaries Stratigraphic column |
| Develop an argument that is consistent with available evidence and uncertainty | Analysis of structural data in deformed rocks | Stereoplot |
| Communicate clearly using written, verbal, and/or visual media (e.g., maps, cross-sections, reports) with discipline-specific terminology appropriate to your audience | Making a geological map, constructing geological cross-sections and a stratigraphic column | Geological map Cross sections Stratigraphic column |
| Synthesize geologic data and integrate with core concepts and skills into a cohesive spatial and temporal scientific interpretation | Synthesis of field data and interpretations of stratigraphic relationships and structures to establish the geological history of an area | Daily summaries |
|  | Interpretation of deformed rocks | Daily summaries Geological map Cross sections |
| Design a field strategy to collect or select data in order to answer a geologic question |  |  |
| Reflect on personal strengths and challenges (e.g. in study design, safety, time management, independent and collaborative work) |  |  |
| Demonstrate behaviours expected of professional geoscientists (e.g., time management, work preparation, collegiality, health and safety, ethics) |  |  |
| Work effectively independently and collaboratively (e.g., commitment, reliability, leadership, open for advice, channels of communication, supportive, inclusive) |  |  |



**Table 1. Learning outcomes from the National Association of Geoscience Teachers (NAGT) for virtual field experiences and for the Esla Mapping Project, indicating the tasks in which they are involved.**

## 2.6 Objectives and assessment

The objectives of the EMP were to produce a report with five assessed sections (Fig. 3). The first one consisted of a daily summary of the field notebook that contained (i) some of the information explicitly provided in the stops, (ii) information inferred from outcrop photographs and sketches such as rock type, age, younging direction, sedimentary structures and tectonic structures, (iii) bedding data calculated from structural contours, (iv) annotated sketches drawn from panoramic photographs, (v) interpretation of depositional environments, (vi) provisional stratigraphic columns, (vii) and a table of geological events. The second section was the production of an A3 geological map from the information provided and inferred. Students were instructed to colour each of the KMZ stops according to their descriptions, from which they had to infer the different lithological units and the nature of their contacts (*i.e.* conformable, unconformable, tectonic), and correlate them throughout the area, in a similar way to the physical mapping exercise. With the aid of topographic contours and GE satellite images, they drew form lines for different lithologies and identified high-angle faults affecting the sequence. All the information was to be transferred to the topographic hard-copy map, where students were asked to draw the contacts of the different identified lithological units, colour them, identify and draw fold axial traces, and draw bedding and younging symbols. The third section consisted of drawing two cross sections in the south and north of the map, choosing appropriate locations normal to the local strike of the lithological units. The fourth section was a stratigraphic column with the defined lithological units, with an estimation of their thickness from the geological map and cross sections, an estimated age based on fossil content, a brief lithological description and an inference of their depositional environment. Finally, the fifth section consisted of plotting bedding plane measurements on a lower-hemisphere equal area stereoplot, and recognising and calculating fold hinges using π pole girdles from selected data.

## 2.7 Duration and implementation of the EMP

After a one hour virtual briefing about the EMP, the students had 20 days before electronic submission of the daily journals, map, cross sections, stereoplots and stratigraphic columns. During this period three online Q & A sessions of one hour each were run by teaching staff, and 6 one hour online clinics were run by postgraduate demonstrators, which mimicked the style and level of contact teaching that would have been carried out in the field. Students were encouraged to work in pairs or groups of three, but it was emphasised that they needed to submit their work individually and that plagiarism would be checked.

## 3 Analysis of the EMP implementation

In this section, results of the implementation of the EMP are described based on a qualitative comparison between the final outcome of the physical and virtual trip in the same area, an evaluation of the student EMP reports, and the analysis of student





feedback through questionnaires, adapting previous fieldtrip evaluations described in the literature (*e.g.* Boyle *et al.*, 2007;

Elkins and Elkins, 2007; Stokes and Boyle, 2009).

### 3.1 Comparison between the outcomes of the physical and virtual trips

The report produced by a 2nd year student in the field during the 2019 campaign was compared with a report produced by the same student via the EMP in 2020. Although the student had gained considerable experience in the time between both reports, this comparison allowed differences in the mapping experiences to be evaluated. The time spent on both trips was similar,

around 8 hours/day during five days. The area covered in the EMP is larger than during the physical trip (10 vs. 6 km²), which results in the recognition of more lithological units (stratigraphic column of 2.2 vs. 0.85 km) and a refinement of the data used for fold axes calculations. However, this came at the expense of the level of detail attained through the EMP, which is lower than what could be achieved in the field, in especially complex areas where geological changes occur at a small spatial scale, such as some exposures of the base of the Esla nappe.

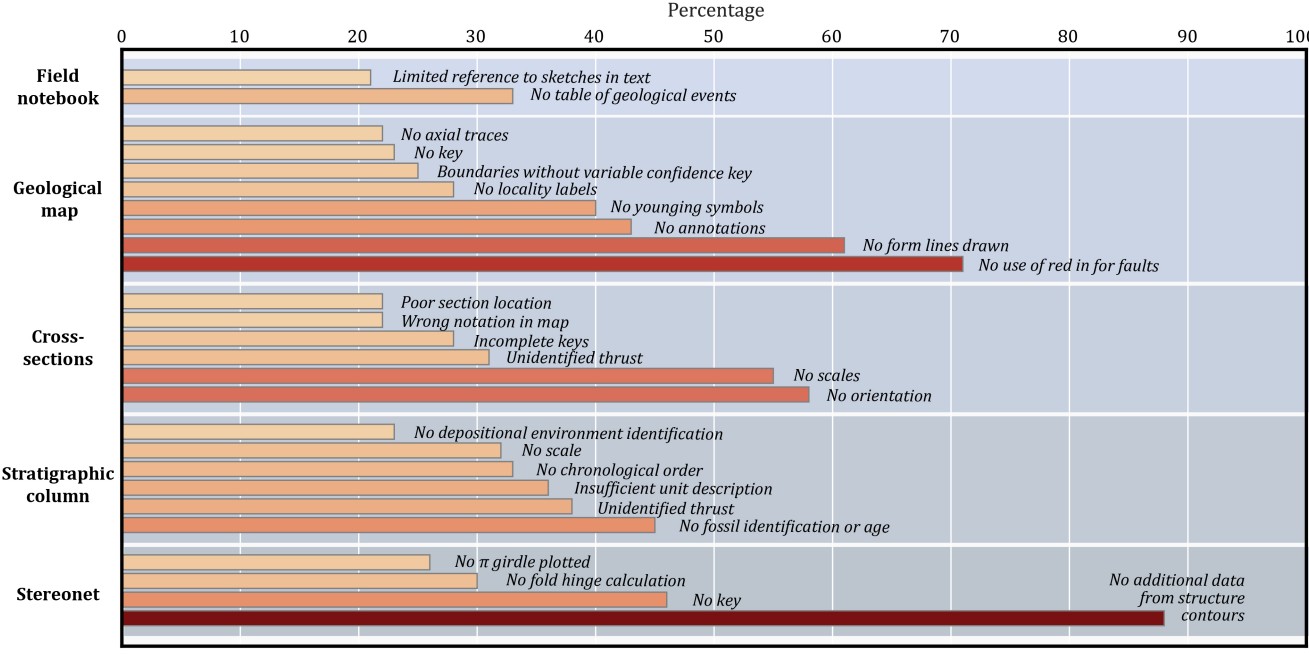


**Figure 4. Most common student mistakes in the different tasks that comprise the EMP. Most of them could be easily rectified with a more comprehensive set of instructions and preparatory lessons.**

### 3.2 Evaluation of student reports

Most students identified and grouped formations appropriately, and mapped geological boundaries in approximately correct

places. The majority also identified the critical Esla thrust. The major weaknesses of the maps, apart from failing to follow good mapping conventions such as using a different colour for structures, and omitting younging symbols or a key, was the lack of form lines on the maps, which inevitably meant that geological boundaries were not realistic with respect to topography





(Fig. 4). The geological summaries were sometimes little more than repetitions of supplied information with no consideration of its significance. Sketches in the summaries were commonly of the photographs supplied, and rarely of landscape views

visible in Google Earth. Cross sections mainly lacked conventional section information such as orientations or one or both vertical and horizontal scales. Stratigraphic columns lacked fossil identification and ages or formation descriptions. The most common weakness of the stereoplots was failure to add data obtained from structure contour measurements, but lack of key, fold hinges and pi-pole girdles were common. The reports produced from EMP obtained lower marks than the reports produced during physical trips in previous years (56% vs. a five-year average of 62%).

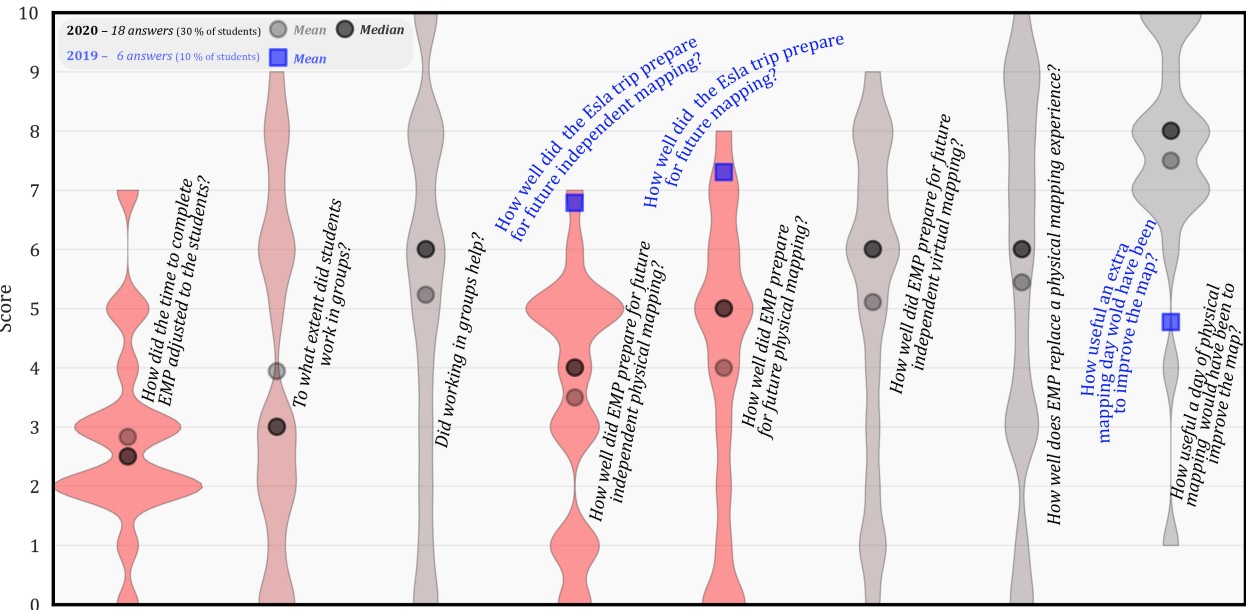

**Figure 5. Violin plots (i.e. two-sided histograms) of the students' answers to a score-based questionnaire (Q2) that surveyed their responses to the EMP. Note the large spread in their answers. In three cases, the same question was posed to previous year students (Q3) that participated in the residential field trip to the Esla Nappe: they are labelled in blue.**

### 3.3 Analysis of student feedback

Students that attended the EMP were given two questionnaires: one at the end of the exercise (Q1), and another one after 8 months (Q2). Students that attended the physical fieldtrip in 2019 were asked to complete a different questionnaire 20 months after their trip (Q3).

The main outcome of the student textbox comments in Q1 and Q2 is in agreement with the results derived from the report assessment. Students felt the need for a more extensive taught section prior to the start of the EMP including a reminder of

geological concepts needed to adequately complete the EMP and a basic crash course in the use of GE (though this was included in the briefing), as well as more detailed instructions on the different tasks and the estimated time they should spent on each of them. The results of the score-based questionnaires are shown in Fig. 5. The spread in student answers shows that there was a range of experiences for the students. While for some of them the experience seems to have been very positive,





others consider their geological skills have not particularly improved as a result of the EMP. There were three common answers

to Q2 and Q3, related to the perceived student ability to undergo future independent mapping campaigns after EMP (Q2) or

physical fieldtrip (Q3). Unsurprisingly, students feel much less prepared to face a mapping project on their own after the EMP

than after the physical trip (3.7/10 vs. 7/10, respectively).

## 4 Discussion

### 4.1 Evaluation of the learning outcomes

The EMP can be considered, overall, a satisfactory experience for the students in developing key geological skills. A

comparison between the continuous digital geological map of Spain (GEODE) from the Spanish Geological Survey (IGME;

Merino-Tomé *et al.*, 2014) and a selected area of the geological map produced by students in the EMP is provided in Fig. 6.

The map attests to success in the students' learning in the EMP of the extrapolation skills necessary for the elaboration of a

geological map. As it would be expected from any geological survey at a 1:10000 scale, it allows the differentiation of

lithologies at the member-scale of officially defined formations.

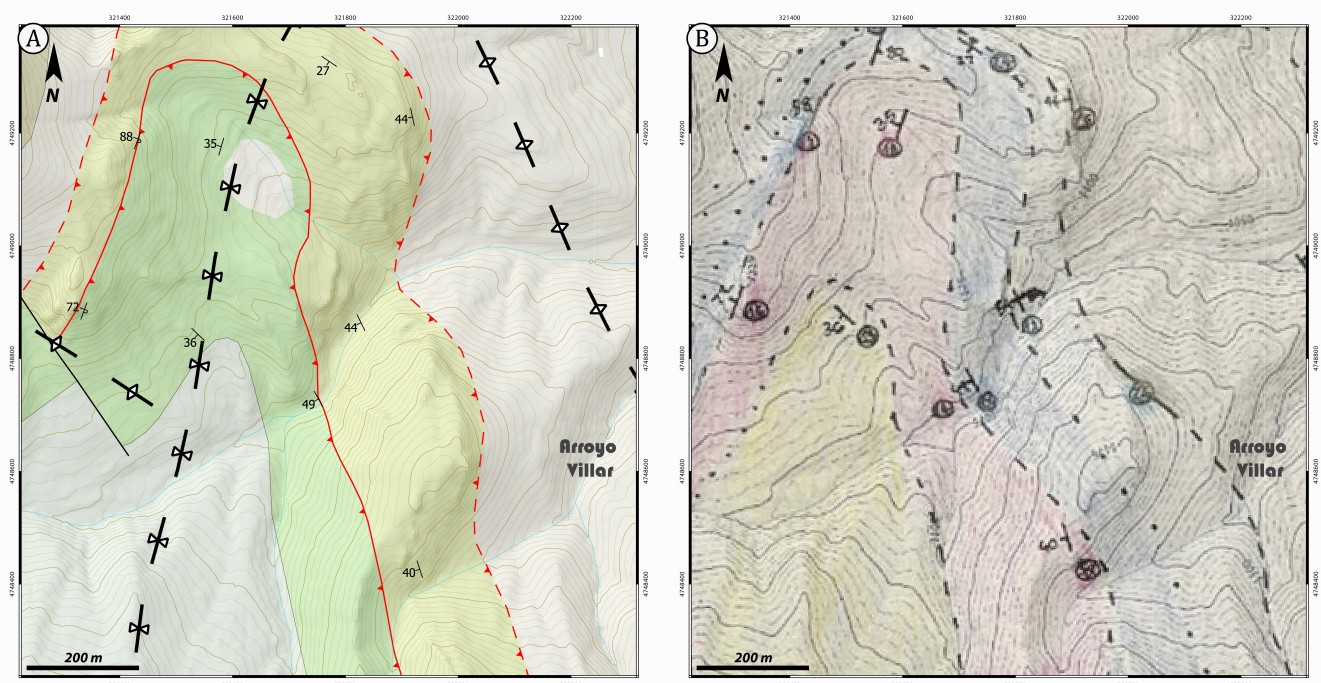

**Figure 6. Detail of the geological map of the Esla Nappe immediately west from the locality of La Velilla de Valdoré. (a) Modified version of the continuous digital geological map of Spain (GEODE) from the Spanish Geological Survey (modified from Merino-Tomé *et al.*, 2014). Bedding data is the same as provided to the students in the EMP. (b) Geological map produced by one of the**

**students as a result of the Esla Mapping Project.**





The learning outcomes were variably met by the students, as attested by the large spread in the grades achieved in the different tasks (Fig. 7). The grades related to the field notebook and the geological map are lower than in previous years of the physical trip (Fig. 8). On the other hand, the grades of the cross sections, stratigraphic columns and stereoplots are in the same range as in previous years. This is not surprising, since these tasks have the lowest degree of field interaction. The lower overall marks

attained by the students doing the virtual project than previous year's physical mapping trips is a cause for concern about the relative effectiveness of the virtual trip in achieving the learning outcomes. Several factors may have played a role. Preparation of students may have been at a lower level prior to the EMP arising from the COVID-19 lockdown and associated decrease in practical training. There may have been less training in digital mapping techniques than in physical mapping. However, the majority of marks were lost in mundane failure to follow standard protocols for preparation of maps, sections, stratigraphic

columns and stereoplots, such as omission of keys and scales. One major factor behind this may be the lack of peer-to-peer learning which is greatly facilitated on a physical trip by contact in the field and in the evening when maps are revised and "inked-in". Another possible explanation is that mapping in the field requires a high level of physical and intellectual engagement by the students, which could foster deep learning but is difficult to fully replicate with a virtual exercise.

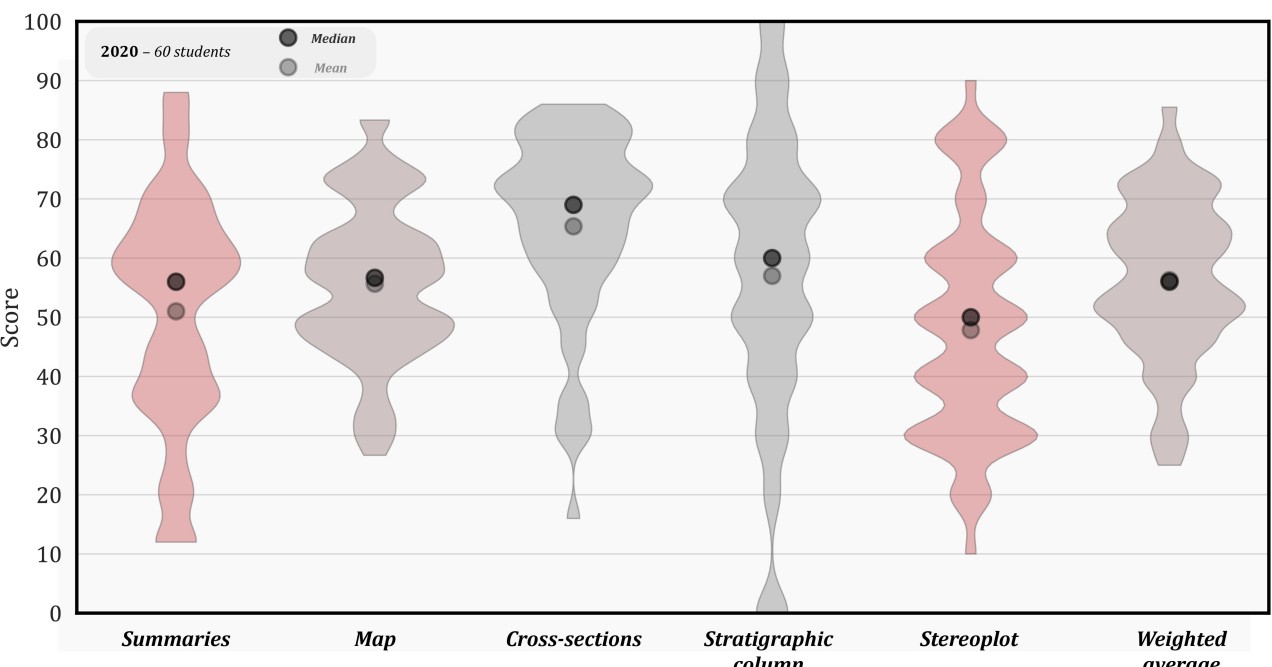

**Figure 7. Violin plots (i.e. two-sided histograms) of the students' grades on the different evaluated tasks. Note the large spread in the grades.**

From the evaluation of student reports and surveys, the implementation of the EMP lacked some areas that would have helped the students in improving their learning outcomes. These largely refer to preparatory teaching and the instruction document. Some students struggled with (i) the drawing and annotation of geological sketches; (ii) the nomenclature of important

geological structures; (iii) the relation between the attitude of lithological units and topography; (iv) the adequate construction





and annotation of a cross section; (v) the construction of stratigraphic columns, and how they are different from sedimentary logs; (vi) the plotting of data on a stereonet and its use to derive structural data such as fold hinges and hinge surfaces; (vii) the basic functioning of GE, and (viii) the taking of document photographs, some of which had low quality largely due to lighting conditions. Although some of this content is addressed in other courses during 1$^{st}$ and 2$^{nd}$ years, students would benefit

from a short revision of each of the problematic areas through online lectures prior to the undertaking of the EMP. The instructions provided for the completion of the EMP were unsatisfactory for some students, who felt uncertain as to the time to be spent on each section of the report, which resulted in some students spending longer hours than others, affecting the quality of the reports and the students' time management. In some tasks, particularly the daily summaries, the students felt doubtful about what was expected to be included, which resulted in a general decrease in the potential grades. During year

2020-2021, students were better briefed on the tasks to be completed, which has apparently resulted in an overall improvement of the grades (Fig. 8).

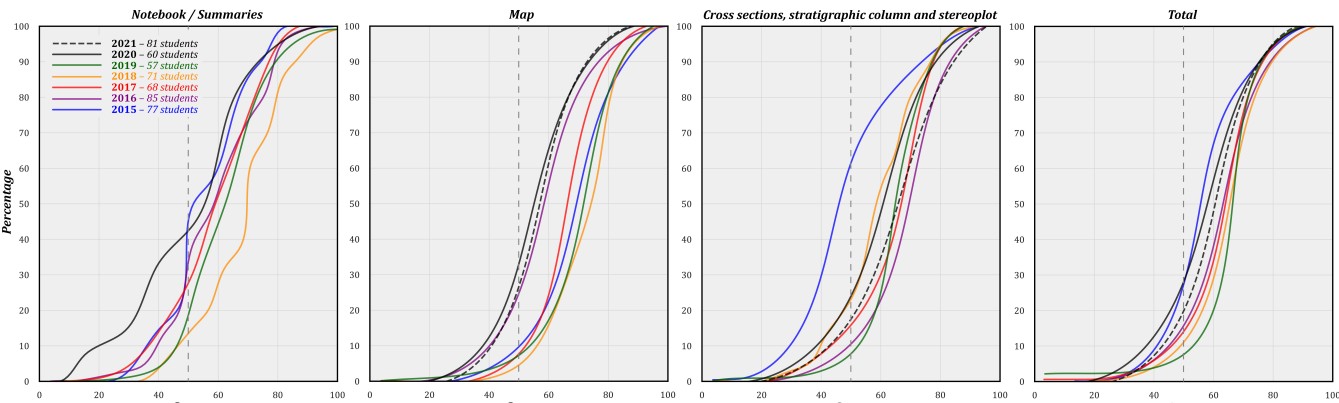

**Figure 8. Cumulative plots of the students' grades on different tasks throughout the years. The grades from the EMP (black colours, years 2020 and 2021) are, in general, lower than previous years'. Note that during 2021 summaries were not evaluated, and were**
**not accounted for in the final grade.**

Some of the above problems are common mistakes for the weaker students in any circumstances, but they may have been made more frequently in the virtual environment due to a lack of confidence and due to the lack of peer-to-peer learning.

Unfortunately, the percentage of students that filled in the questionnaires was low, which hinders the extraction of truly representative conclusions. This is a recurrent issue when assessing educational outcomes, especially in the case of small

student groups (e.g. Chapman and Joines, 2017). In the case of a course with 60 students, as is the case for 2$^{nd}$ year in the Geology BSc at Cardiff University, Nulty (2008) suggests that a minimum response rate of 31 % is required to ensure the representativeness of the survey answers. In our case, the response rate for the questionnaires was, respectively, 21 %, 30 % and 10 %.

An important consideration is that the EMP was only possible thanks to the wealth of data previously collected by the authors

over the years, whose extent covers the entire mapping area, thus making possible a realistic fieldtrip. In addition to the geological data, the abundant photographs from the area have allowed to convey the outcrop descriptions adequately to the





students, and to train the students' ability to draw realistic field sketches from the same area as if they were there physically. In our view, this data and material abundance is a crucial bottleneck decisive for a successful implementation of a similar virtual mapping exercise in any other area.

**4.2 Advantages and disadvantages of the EMP**

Despite its success in training numerous field geological skills, when undertaking the EMP the students do not gather the geological data themselves. Students are not trained in procedures inherent to field geology, such as observing the rocks at the outcrop scale, extracting samples to be inspected with a hand lens, identifying their lithology, using a compass to measure strike and dip of bedding and/or any other structures, looking for deformation and sedimentary structures and polarity

indicators, or looking for fossils and identifying them. Students are also not trained in more logistical issues such as the ability to make real-time itinerary modifications, itinerary planning for future days based on available data, and crucially, terrain navigation and self-location in the field. These are all fundamental skills that a field geologist needs to develop, and are in our view very difficult to replicate adequately in a virtual environment. The lack of social interaction between students, fundamental to foster peer-learning, has likely influenced the lower grades of the EMP in comparison with previous years.

These are important drawbacks of the EMP as a substitutive activity for a physical residential fieldtrip, as is true for any other virtual fieldwork exercise (e.g. Hurst, 1998; Arrowsmith et al., 2005).

However, the students have met the learning outcomes of the EMP to a similar extent as in previous years, as attested by their grades in the different tasks (Figure 8). Furthermore, the EMP allows students to learn skills not implemented in the physical trip in previous years. These include (i) fieldwork planning using aerial imagery, (ii) drawing of form lines, (iii) visualising

the three-dimensional interactions of the geological structure with the topographic surface in a virtual globe environment (Google Earth), or (iv) making geological interpretations based on observations made by other geologists, a fundamental ability for professional geologists in an increasingly time-deprived society and the growing multidisciplinary approaches in industry and academia.

The EMP provides a curricular opportunity for students who are unable to attend residential physical fieldtrips for various

reasons, including students with mobility issues or those responsible for a person reliant on care. There is an increasing demand for inclusive learning and equality of learning opportunities for these students, which have traditionally been neglected (Hall et al., 2004; Stokes et al., 2012; Atchison et al., 2019). The EMP provides these students with an exercise where important geological field skills are learnt, and the implementation of the activity can greatly benefit them.

Finally, the EMP is easily adaptable to new circumstances and learning needs. For example, during the 2020/2021 academic

year, the fourth day of the EMP was deleted, leaving it with a total of four days instead of five. This preserves the learning outcomes unaltered, while at the same time relieving the time-pressure imposed on the students in the last weeks of the academic years, when the activity is carried out. Furthermore, once the fieldtrip workflow has been implemented and the results tested, the format can be applied to different field areas where abundant data is available to produce a similar exercise.



### 4.3 Possible improvements

Most student mistakes could be minimised in the future by even more emphasis on the prescriptive list of what needs to be included in each of the evaluated sections, and with further preparatory lessons on basic geological skills such as the generation of geological sketches, mapping symbology, the information recorded in a cross-section, the production of stratigraphic columns, and the plotting of data on a stereonet. These are all basic skills that the students practised throughout their 1st and 2nd year, but the EMP experience has shown that they still have not interiorised them, at least not well enough to work

independently without the support of lecturers and assistants and, critically, peer-to-peer learning.

There are other aspects of the trip that could be improved for future runs of the EMP. One addition to the fieldtrip would be to promote more involvement of the students in recognising some of the lithologies present in the area. This could be implemented virtually in some of the stops, for example by showing a thin section scan or a series of microphotographs taken with an optical microscope. This approach, in combination with outcrop photographs, could improve at the same time their petrographic skills

as well as their linking with outcrop-scale sedimentary features (*e.g.* Marshall and Higley, 2021). Another improvement could be allowing the students to plan their own daily itineraries based on the collected data in previous days, so that they can decide where to focus their efforts based on which areas look more promising, just as in physical fieldwork. This would foster their ability to manage their available time in the field in the future. This could be implemented, for example, by generating a series of stops that they could link at their will, generating many possible itineraries. When choosing the right combination, students

would get the sense of the structure in a few stops. If they choose poorly, they would soon realise that there were better combinations to suit their mapping needs, and react accordingly. A similar method has been implemented by Mahan *et al.* (2021), who give the students the chance to choose their mapping stations.

### 5 Conclusions

Physical field geology is an irreplaceable curricular activity in any Earth Science undergraduate and postgraduate programme,

and in particular Geology. In a situation (even prior to COVID-19) where universities are subjected to increasing financial pressure, the number of field-based geoscience curricular activities is progressively decreasing (e.g. Boyle et al., 2007, Mullens et al., 2012). This has caused alarm among geoscientists, who advocate for the maintenance of physical fieldtrips (e.g. Boyle et al., 2007; Butler, 2008). Our experience with the EMP agrees with this view: the virtual fieldtrip experience, though satisfactory as a substitute experience in a period of crisis and effective in the training of numerous skills, cannot replace a

physical mapping course in the field (e.g. Spicer and Stratford, 2001). Nonetheless, we consider that the EMP has inherent benefits, and some of its parts could be implemented in combination with the physical trip in the future. Some of the activities of the EMP, such as form line drawing and inspection of the interaction between geological structure and topography, could provide an outstanding introduction to the mapping area in preparation for a physical trip. This would allow for a better time-management of the trip, with the targeting of especially problematic areas that require a more detailed attention in the field

(e.g. where aerial imagery is not conclusive, in lithological contacts, fold hinges, etc.). This positive combination of virtual

and physical trips is also supported by other educators after their analysis of student learning outcomes, both prior and during the COVID-19 pandemic (e.g. Stainfield et al., 2000; Arrowsmith et al., 2005; Granshaw and Duggan-Haas, 2012; Litherland and Stott, 2012; Cliffe, 2017; Rotzien et al., 2021; Toy et al., 2021; Bond and Cawood, 2021, Evelpidou et al., 2021).

**Data availability**

The KMZ files, outcrop descriptions, and photographs, are all stored in the GitHub repository. It can be accessed through the following link: https://github.com/EslaUnit/EslaMappingProject. The references used to obtain fossil images and information, as well as stratigraphic logs, that are not mentioned in the text, are listed in a separate supplementary document. The topographic map supplied to the students can also be found in the supplementary material. The fossil guide and the set of instructions provided to the students is also provided as supplementary material.

**Author contribution**

TGB, DMB and MIdPA designed the EMP and provided real geological data and photographs for the exercise. TGB designed the activities to be performed by the students, and produced the EMP manual for the students. TGB, DMB and other staff implemented the EMP at Cardiff University. GEG conducted the student surveys and analysed the students' marks. LC compiled the fossil guide for the students. TGB outlined the manuscript structure, which was expanded upon by MIdPA with 395 contributions from the co-authors.

**Competing interests**

The authors declare that they have no conflict of interest.

**Special issue statement**

This manuscript has been prepared for submission to Solid Earth to be part of the special issue Virtual geoscience education 400 resources (SE/GC inter-journal SI).

**Acknowledgements**

This contribution results from a collaboration between the Cardiff University and the University of Oviedo. It has been possible owing to the research cooperation between researchers from different countries, economically crystallised through the research project PETROCANTÁBRICA (MINECO-18-CGL2017-86487-P) supported by the Ministry of Economy and 405 Competitiveness of Spain (MINECO). M.I. de Paz-Álvarez acknowledges a pre-doctoral contract from the FPU program of



the Ministry of Education of Spain. The staff of Cardiff University involved in the Esla trip is thanked for their support and implementation of the EMP. Students of 2nd and 3rd year from the Geology BSc of the Cardiff University are thanked for their participation in the survey.

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
