# Peer review of "Virtual fieldtrip to the Esla Nappe (Cantabrian Zone, NW Spain): delivering traditional geological mapping skills remotely using real data"

_Solid Earth, 2021_

## Referee Comment (RC2)

**Review of de Paz-Alvarez et al. 2021- Virtual field trip…**

Jaime Toro, West Virginia University

I enjoyed reading the paper by de Paz-Alvarez et al. It is well written, well organized and well illustrated. However, I found that the authors were not sufficiently critical of the results of their experiment. When they asked the students to rate how well the Virtual Field Trip prepared them to face a real mapping project on their own, and their answer was 3.7/10. Frankly, this is a failing grade. De Paz-Alvarez et al. should conclude that this was not the right approach. Notice that that example of student work presented in Fig. 3 is very nice looking, but (according to the authors) the descriptions are copied from the lithologic tags illustrated in Fig. 2, and the cross section is wrong! The main structure should be a recumbent syncline (Fig. 6A), not an anticline. So, what did that student really learn?

Covid posed a dilemma to all of us who teach field geology. We could either attempt to reproduce the field experience virtually (which is the approach taken by de Paz-Alvarez et al.) or we could use the opportunity to do something entirely new that fulfilled some of the same learning outcomes as the field experience. The problem with the virtual field trip approach is that it becomes fundamentally a cookbook exercise, an artificial puzzle. It can teach the students certain skills, such as keeping neat notes and drawing cross sections at their desk. But it will not prepare them for real field work any more than playing *Fortnite* would prepare you to take part in the landing in Normandy. The point of a field experience is precisely learning to deal with those elements that the authors acknowledge to be missing from the virtual field trip: route-finding, orienteering, the size of the world, its slopes, its messy outcrops, its hard rocks that you need to hit with a hammer, its prickly bushes, its noisy complexity, its changing weather, plus our annoying companions,  etc. In other words, field work is about experiencing the difference between a neat sketch on a piece of paper and an imperfect outcrop in the woods.

 I think the preferable approach is not do a virtual field trip but to teach the students how to use the tools that a professional would use to solve a real geological problem remotely. In other words: teach them to use remote sensing data, GIS, geodetic data, potential field data, or to interpret geological data collected by others for a realistic reason (not just to make a virtual field trip). In order to apply this to the Esla Nappe project the students would need to learn how to use QGIS, a step considered by the authors but rejected as impractical. Teach them how to use slope and aspect of a DEM to collect strike and dip data. Teach them how to draw the contacts in the GIS. Those are examples of skills that are applicable in a professional setting. Copying data from somebody else's field stops has very limited value.

I recommend that the authors revise their abstract, conclusions and discussion to more clearly acknowledge the limited success of virtual field trips in teaching field skills. This was not a successful experiment; it is foolish to promote it as such.

**Specific Comments on the Manuscript**

103 EMP- Please spell it out: Esla Mapping Project or Esla Project. Non-standard acronyms are the bane of scientific writing. They save virtually no space but make papers hard to read.

Fig. 2 Insets are labelled A, B, C but the caption says a,b,c . Be consistent!

195 I think GE means General Electric.

Figure 4- What is the number of students? N= ? I notice that you did not grade the cross sections for structural correctness. That seems strange. Isn't that the point of a cross section?

Figure 5. N=? Some of those questions are awfully similar to each other. If I were taking this poll, I would have been confused.

Figure 7 N=?

260 The evidence presented in line 257 shows that the virtual field trip did not accomplish what it set out to do.  Delete this line.

Figure 6. The student did not map the fault, did not recognize the overturned beds and did not identify the syncline. I don't see how you can claim that this is a satisfactory result.

321 I do not think your paper demonstrated that virtual mapping trains "numerous field geological skills". It does train some office skills.

---

## Author Response (AR1)

**REVIEWER 1 – Lidia Lonergan**

In this section we address the comments provided by Lidia Lonergan other than single word corrections in the text, which have been modified accordingly.

**Specific Comments:**

- Comment: At several points in the paper it states or infers that 5 days/ 40 hours were allocated to the virtual course -to replicate 8x 5 field days; elsewhere it is stated that students had 20 days before electronic submission (p. 10 line 206); it would help to clarify how long the students were expected to spend on the virtual fieldwork (e.g. 5 consecutive timetabled days?) and how long was allocated to 'writing up' results, including drawing sections, finalizing map etc.

  Answer: Overall, the students had 20 days to produce their reports. But the amount of work expected from them was in the order of 40 hours, which they could freely distribute among the 20 days provided. Ideally, the intention was for the students to spend 8 hours for 5 days in order to reproduce as closely as possible the workflow used in the residential fieldtrip. Nonetheless, given the exceptional circumstances, and being unaware of the personal circumstances of the students during lockdown, it was decided that they should be given an ample deadline so that they could work on the project whenever they chose to.
  There was not a clear instruction on the time allocated to producing the report in itself, since they were expected to do part of it daily (e.g. completing a field notebook and the geological map). Once all data was reviewed, then they had to do the rest: stratigraphic column, cross-sections and stereonet. They were expected to do everything in 40 hours. Nonetheless, the reviewer's comment is very helpful for future runs of the exercise in case the reports need to be as thorough as during the years 2020 and 2021: it is a good point that students should have at least an approximate notion of the time available for that purpose.

  Changes: In section 2.7, the following sentences have been added for clarification: *Students were expected to work on the project for about 40 hours, which is a similar amount as during previous years' residential fieldtrip. The deadline of 20 days was given in order to ensure all students had enough time to adequately complete the exercise bearing in mind their different personal circumstances during lockdown.*

**Technical corrections**

- (L 29) Replace 'transversal'; should it be 'transferrable'?
  To our knowledge, those are synonyms, but a quick online survey shows that indeed, transferable is more common. We have modified it in the text.
- (L 73) Replace 'these trainings'  training not generally used in plural; 'these approaches' or these courses might be appropriate
  We have modified it in the text.
- (L 95) 'sub-greenschist' add hyphen.
  We have modified it in the text.
- (L 299-300) 'During year 2020-2021, students….."  Rewrite – during the 2020-21 version of the course
  We have rewritten that sentence.
- (L 318) "In our view, this data and material abundance is a crucial bottleneck decisive for a successful implementation"   Rewrite – I don't fully understand.
  We have modified the sentence as follows: *In our view, this data and material abundance is a decisive factor for a successful implementation of a similar virtual mapping exercise in any other area.*

Figure 5 – Violin plots. Is label on first histogram correct? "How did the time to complete EMP adjusted to the students?" Likewise blue label on final histogram needs word order changing and spelling correction- "How useful an extra mapping day wold have been to improve the map?" and grey label on same histogram (word order).

We have modified the first label: *Was the available time enough to complete the EMP successfully?*
The final blue label: *To what extent would an extra mapping day have improved the map?*
The final grey label: *To what extent would a day of physical mapping have improved the map?*

**REVIEWER 2 – Jaime Toro**

In this section we address the comments provided by Jaime Toro.

**Overall review**

- *Comment: I found that the authors were not sufficiently critical of the results of their experiment. When they asked the students to rate how well the Virtual Field Trip prepared them to face a real mapping project on their own, and their answer was 3.7/10. Frankly, this is a failing grade. De Paz-Alvarez et al. should conclude that this was not the right approach.*
  The reviewer suggests that the low grade given by the students to the question 'How well the EMP prepared students for a real independent mapping project' indicates failure of the EMP. There are several issues at play behind this critique:
  Firstly, this "failing grade" was a response by students to one specific question about the project, which cannot be construed as a measure of the success/failure of the exercise. Questionnaire responses, however negative, are not the same as an evaluation of the learning outcomes. It is disappointing that the students did not feel more prepared, but they have very limited field experience on which to make this assessment. With further reflection, this was not even a particularly useful question to ask.
  Secondly, we whole-heartedly agree with the sentiments of the reviewer that a virtual exercise will never come close to a real mapping experience, and it is clearly stated as the authors' opinion throughout the manuscript. This exercise was developed under circumstances where a real mapping experience was not possible due to the COVID-related travel and social restrictions. It is natural that students do not feel as prepared to undertake an independent mapping project after the EMP than after a physical trip. Both activities are simply not comparable. In our view the EMP exercise does just what the reviewer recommends: "…*to do something entirely new that fulfilled some of the same learning outcomes as the field experience*." We aimed to achieve as many as possible of the outcomes of the field experience, but obviously there are fundamental aspects, such as those highlighted by the reviewer, that can never be achieved through a virtual exercise.
  Thirdly, the rate of response of the students to the questionnaires was 30 %, which is unfortunately low. The low response rate does qualify the significance that can be attached to the questionnaire results.

- *Comment: Notice that that example of student work presented in Fig. 3 is very nice looking, but (according to the authors) the descriptions are copied from the lithologic tags illustrated in Fig. 2, and the cross section is structurally wrong! The main structure should be a recumbent syncline (Fig. 6A), not an anticline. So, what did that student really learn?*
  The reviewer uses as an example of the failure of the exercise a supposedly hopelessly wrong cross-section where an anticline is drawn instead of a syncline. The truth is, the cross-section was drawn in a different location than that shown in Figure 6, a location where it crosses the Pardominos anticline that generates the Valdoré tectonic window (*e.g.* Alonso, 1987). This structure is easily seen in the

central part of Figure 1 striking NW – SE. This is a misunderstanding caused by the authors' failure to locate the cross-section in the manuscript.

*Changes:* The caption of figure 6 has been modified to clarify that the cross section is not drawn across the fold in Fig. 6. A square-box has been added to Fig. 1B to highlight the location of Figure 2A and 6. The caption of figure 3B has been modified stating the name of the anticline drawn as known in the regional geology literature.

- *Comment: We could use the opportunity to do something entirely new that fulfilled some of the same learning outcomes as the field experience. The problem with the virtual field trip approach is that it becomes fundamentally a cookbook exercise, an artificial puzzle.*
  The EMP does attempt to fulfil some of the same learning outcomes as the field experience: this is emphasised in lines 80-84 of the original manuscript. The EMP exercise also attempts to go one step beyond than previous virtual field trips, making the experience as close as possible to a physical trip from a methodological point of view. It is not a cookbook exercise, at least no more than a real residential fieldtrip, in the sense that students are also asked to routinely perform a series of tasks in that case. Of course, many fundamental skills cannot be developed in our exercise, as acknowledged in the manuscript. It is very important to be able to identify rocks, but so is the ability to synthesize lithological and fossil descriptions into identifiable, coherent lithostratigraphical units. This skill is a focus of the EMP.

- *Comment: I think the preferable approach is not do a virtual field trip but to teach the students how to use the tools that a professional would use to solve a real geological problem remotely. In other words: teach them to use remote sensing data, GIS, geodetic data, potential field data, or to interpret geological data collected by others for a realistic reason (not just to make a virtual field trip).*
  In our view, the technology used to produce a map is much less important than the mapping skills. Thus, learning GIS does not guarantee success in the learning of geological mapping: quite the contrary, if learning GIS undercuts time spent on learning geological concepts.
  The geological data used in the EMP was collected with a completely different objective than making a virtual field trip: part of it was collected during years of teaching in physical trips (that is, for the sake of education), and the rest was collected during a PhD project focussed on the deformation at the base of the Cantabrian nappes (for the sake of research). It so happened that combining both data, the authors were in a very good position for producing something like the EMP, whose development was incidental to the data collection.

- *Comment: In order to apply this to the Esla Nappe project the students would need to learn how to use QGIS, a step considered by the authors but rejected as impractical. I think yuo need to bite the bullet and teach them how to use slope and aspect of a DEM to collect strike and dip data. Teach them how to draw the contacts in the GIS. Those are examples of skills that are applicable in a professional setting. Copying data from somebody else's field stops has very limited value.*
  Students were required to draw geological contacts in Google Earth, making use of the satellite images, in order to have a good three-dimensional view of the area and to better translate them onto the topographic map. As for using the slope of DEM to collect strike and dip, students were specifically asked to do so in some of the stops where provided data was purposefully missing strike and dip orientations, as we explain in sections 2.6, 3.1 and Figure 4. They were taught in the preparatory briefing how to construct structural contours and to extract strike and dip from Google Earth. Nonetheless, perhaps we should emphasize this point more in the manuscript.

QGIS was considered as the tool for the EMP but the students lacked the necessary GIS skills to use QGIS and since the course was designed during lockdown, it was not viable to ask them to learn the basics principles of QGIS and produce a geological map at the same time.

With respect with two last sentences above, we emphasize that they did not simply copy the supplied lithological descriptions, but they had to build their lithostratigraphic column based on these. It would have been all too easy for students to consider, say, all shale outcrops part of the same unit and have produced an erroneous map, but that was not usually the case. Note that they were not provided with the established stratigraphic succession in the area, and that they had to derive that from objective data. This, in our view, has a lot of methodological value.

Changes: We have added a more clear description in section 2.6 of the requirement for the students to produce their own bedding orientation from structural contours derived from lithological contacts and the provided topographic contours.

- *Comment: I recommend that the authors revise their abstract, conclusions and discussion to more clearly acknowledge the limited success of virtual field trips in teaching field skills. This was not a successful experiment; it is foolish to promote it as such.*
  In the original manuscript, we clearly state that we are not advocating for changing mapping courses from physical to virtual formats. In three places (Abstract, lines 21-22; section 4.1, lines 321-331; Conclusions, 373-375) we recognise that a virtual field trip cannot replace a physical trip, due to a variety of factors which are detailed in the text.

  We do not consider the EMP a failed experiment (otherwise we wouldn't have sent this manuscript). We acknowledge that the EMP has serious limitations when it comes to the training pure field skills such as rock identification, field navigation and orienteering, decision-making. Notwithstanding this, in our view the EMP helps to strengthen other important mapping skills which are essential to the fieldwork experience, which are often delivered during the evening in physical trips: analysing the data, drawing conclusions from the observations, envisioning the relation between the different lithologies, establishing a lithostratigraphic column, adequately translating the geological contacts to the map, choosing the location and orientation of cross-sections before drawing them. These constitute non-field methodologies that are also important in the development of a geologically sound map. Training in this part of the methodology is delivered in the EMP: we focussed on what can be achieved virtually, and not on what simply cannot be accomplished if not in the field.

**Specific comments**

- 103 EMP- Please spell it out: Esla Mapping Project or Esla Project. Non-standard acronyms are the bane of scientific writing. They save virtually no space but make papers hard to read.
  We have modified the acronyms throughout the manuscript.
- Fig. 2 Insets are labelled A, B, C but the caption says a,b,c . Be consistent!
  We have modified figure caption.
- 195 I think GE means General Electric.
  We have omitted Google Earth acronyms.
- Figure 4- What is the number of students? N= ? I notice that you did not grade the cross sections for structural correctness. That seems strange. Isn't that the point of a cross section?
  The number of students was displayed in Fig. 8 and in the text (line 310), and it is 60. Nonetheless, figure 4 would be more complete if stated directly on the figure caption, and so we have added it. The structural correctness of the section was evaluated under several different categories, so that it was not possible to isolate specific mistakes in this category.

- Figure 5. N=? Some of those questions are awfully similar to each other. If I were taking this poll, I would have been confused.

  n and their proportion to N is stated in the figure legend. Nonetheless, we have added it to the figure caption as well. Although similar in their form, the questions target different topics: i.e. students may feel prepared to undertake a future supervised mapping campaign, but not necessarily an independent one. The reason for this question is that in the UK, students are required to undertake an independent mapping project as a major component of their degree.

- Figure 7 N=?

  In this case, N = 60 is clearly stated in the figure legend.

- 260 The evidence presented in line 257 shows that the virtual field trip did not accomplish what it set out to do. Delete this line.

  The EMP did not set out to establish that the virtual field trip achieved the same as a physical trip, but to offer a teaching alternative in a very specific set of circumstances where no other option was available (i.e. COVID19-related social and traveling restrictions). This line simply expresses that the virtual trip was not perceived by students as useful as a physical trip, which is not surprising and was foreseen even before the exercise implementation.

- Figure 6. The student did not map the fault, did not recognize the overturned beds and did not identify the syncline. I don't see how you can claim that this is a satisfactory result.

  Many students did not adequately represent the fault with its correct symbology on the map, but that does not mean that they did not interpret the contact correctly. In particular, the thrust fault is correctly marked as such in the stratigraphic log of Fig. 3A, displayed in a tectono-stratigraphic order, even though is not adequately represented in the map.

  As for the overturned beds, many students failed to represent that symbology, but that is also the case in physical mapping courses, so this is part of their learning process.

  Finally, with respect to the syncline in the map and the anticline in the cross section, we explained in our previous comments how this is a misunderstanding caused by the authors' failure to clearly state that the cross section was not constructed through this syncline, but through the more eastern anticline. Thus, the cross section is actually correct. This is now specified in figure 6 caption.

- 321 I do not think your paper demonstrated that virtual mapping trains "numerous field geological skills". It does train some office skills.

  We do not consider that field geological skills are restricted to rock identification, dip measurement, and terrain navigation. Constructing a geological map and dividing a lithological succession in coherent stratigraphic units, based on field data, is in our view a field skill independently of where the process takes place.